# Structural Determinants and Their Role in Cyanobacterial Morphogenesis

**DOI:** 10.3390/life10120355

**Published:** 2020-12-17

**Authors:** Benjamin L. Springstein, Dennis J. Nürnberg, Gregor L. Weiss, Martin Pilhofer, Karina Stucken

**Affiliations:** 1Department of Microbiology, Blavatnik Institute, Harvard Medical School, Boston, MA 02115, USA; 2Department of Physics, Biophysics and Biochemistry of Photosynthetic Organisms, Freie Universität Berlin, 14195 Berlin, Germany; dennis.nuernberg@fu-berlin.de; 3Department of Biology, Institute of Molecular Biology & Biophysics, ETH Zürich, 8092 Zürich, Switzerland; gregor.weiss@mol.biol.ethz.ch (G.L.W.); pilhofer@mol.biol.ethz.ch (M.P.); 4Department of Food Engineering, Universidad de La Serena, La Serena 1720010, Chile; kstucken@userena.cl

**Keywords:** cyanobacteria, morphology, cell division, cell shape, cytoskeleton, FtsZ, MreB, IF proteins

## Abstract

Cells have to erect and sustain an organized and dynamically adaptable structure for an efficient mode of operation that allows drastic morphological changes during cell growth and cell division. These manifold tasks are complied by the so-called cytoskeleton and its associated proteins. In bacteria, FtsZ and MreB, the bacterial homologs to tubulin and actin, respectively, as well as coiled-coil-rich proteins of intermediate filament (IF)-like function to fulfil these tasks. Despite generally being characterized as Gram-negative, cyanobacteria have a remarkably thick peptidoglycan layer and possess Gram-positive-specific cell division proteins such as SepF and DivIVA-like proteins, besides Gram-negative and cyanobacterial-specific cell division proteins like MinE, SepI, ZipN (Ftn2) and ZipS (Ftn6). The diversity of cellular morphologies and cell growth strategies in cyanobacteria could therefore be the result of additional unidentified structural determinants such as cytoskeletal proteins. In this article, we review the current advances in the understanding of the cyanobacterial cell shape, cell division and cell growth.

## 1. Introduction

One of the major perquisites of cellular functions is a structured and coordinated internal organization. Cells have to build and sustain or, if appropriate, modify their shape, which allows them to rapidly change their behavior in response to external factors. During different life cycle stages, such as cell growth, cell division or cell differentiation, internal structures must dynamically adapt to the current requirements. In eukaryotes, these manifold tasks are fulfilled by the cytoskeleton: proteinaceous polymers that assemble into stable or dynamic filaments or tubules in vivo and in vitro. The eukaryotic cytoskeleton is historically divided into three classes: the actin filaments (consisting of actin monomers), the microtubules (consisting of tubulin subunits) and the intermediate filaments (IFs), although other cytoskeletal classes have been identified in recent years [1,2]. Only the collaborative work of all three cytoskeletal systems enables proper cell mechanics [3]. The long-lasting dogma that prokaryotes, based on their simple cell shapes, do not require cytoskeletal elements was finally abolished by the discovery of FtsZ, a prokaryotic tubulin homolog [4,5,6] and MreB, a bacterial actin homolog [7,8]. These discoveries started an intense search for other cytoskeletal proteins in bacteria and archaea which finally led to the identification of bacterial IF-like proteins such as Crescentin from *Caulobacter crescentus* [9] and even bacterial-specific cytoskeletal protein classes, including bactofilins [10]. Constant influx of new findings finally established that numerous prokaryotic cellular functions, including cell division, cell elongation or bacterial microcompartment segregation are governed by the prokaryotic cytoskeleton (reviewed by [11,12]).

Cyanobacteria are today’s only known prokaryotes capable of performing oxygenic photosynthesis. Based on the presence of an outer membrane, cyanobacteria are generally considered Gram-negative bacteria. However, unlike other Gram-negative bacteria, cyanobacteria contain an unusually thick peptidoglycan (PG) layer between the inner and outer membrane, thus containing features of both Gram phenotypes [13,14,15]. Additionally, the degree of PG crosslinking is much higher in cyanobacteria than in other Gram-negative bacteria, although teichoic acids, typically present in Gram-positive bacteria, are absent [16]. The processes of PG biosynthesis and the proteinaceous components involved in the composition of the cyanobacterial PG were previously reviewed [17] and will not be part of this review.

While Cyanobacteria are monophyletic [18], their cellular morphologies are extremely diverse and range from unicellular species to complex cell-differentiating, multicellular species. Based on this observation, cyanobacteria have been classically divided into five subsections [19]. Subsection I cyanobacteria (*Chroococcales*) are unicellular and divide by binary fission or budding, whereas subsection II cyanobacteria (*Pleurocapsales*) are also unicellular but can undergo multiple fission events, giving rise to many small daughter cells termed baeocytes. Subsection III comprises multicellular, non-cell differentiating cyanobacteria (*Oscillatoriales*) and subsection IV and V cyanobacteria (*Nostocales* and *Stigonematales*) are multicellular, cell differentiating cyanobacteria that form specialized cell types in the absence of combined nitrogen (heterocysts), during unfavorable conditions (akinetes) or to spread and initiate symbiosis (hormogonia). Whereas subsections III and IV form linear cell filaments (termed trichomes) that are surrounded by a common sheath, subsection V can produce lateral branches and/or divide in multiple planes, establishing multiseriate trichomes [19]. Considering this complex morphology, it was postulated that certain subsection V-specific (cytoskeletal) proteins could be responsible for this phenotype. However, no specific gene was identified whose distribution was specifically correlated with the cell morphology among different cyanobacterial subsections [20,21]. Therefore, it seems more likely that differential expression of cell growth and division genes rather than the presence or absence of a single gene is responsible for the cyanobacterial morphological diversity [20,22]. In the heterocystous cyanobacterium *Anabaena* sp. PCC 7120 (hereafter *Anabaena*), the multicellular shape is strictly dependent on cell-cell communication through gated proteinaceous complexes, termed septal junctions, which resemble eukaryotic gap junctions [23] and allow the diffusion of small regulators and metabolites such as sucrose [24]. Septal junctions pierce through the nanopores in the septal PG mesh, which are drilled by AmiC amidases [25]. Additionally, isolated PG sacculi suggest that the PG mesh from neighboring cells is connected, allowing the isolation of seemingly multicellular PG sacculi [25,26]. Several components are putatively involved in the function, formation and integrity of septal junctions, including septal-localized proteins such as SepJ (formerly also known as FraG [27]), SepI, FraC and FraD as well as AmiC1/2/3 and the PG-binding protein SjcF1, but only FraD was unambiguously shown to be a direct component of the septal junctions [23]. For reviews on cell–cell communication and the general multicellular nature of *Anabaena* see [28,29,30]. Here we will review the structural and environmental determinants of cyanobacterial shape, division and growth, focusing on the role of cytoskeletal proteins.

## 2. How Do Cyanobacteria Modify Their Cell Shape?

### 2.1. Morphology and Environmental Cues

Cyanobacteria show a high degree of morphological diversity and can undergo a variety of cellular differentiation processes in order to adapt to certain environmental conditions. This helps them thrive in almost every habitat on Earth, ranging from freshwater to marine and terrestrial habitats, including even symbiotic interactions [31]. One factor which can drive morphological changes in cyanobacteria is light.

As cyanobacteria are bacteria that use light to fuel their energy-producing photosynthetic machinery they depend on perceiving light in order to optimize their response and to avoid harmful light that could result in the formation of reactive oxygen species (ROS) and subsequently in their death (reviewed by [32]). Optimal light conditions may be defined by quantity (irradiance), duration (day-night cycle) and wavelength (i.e., the color of light). The photosynthetically useable light range of the solar spectrum is generally referred to as PAR (photosynthetically active radiation), but some cyanobacteria may expand on PAR by not only absorbing in the visible spectrum, but also the near-infrared light spectrum. This employs a variety of chlorophylls and allows phototrophic growth up to a wavelength of 750 nm (reviewed by [33]). To sense the light across this range of wavelengths, cyanobacteria possess various photoreceptors of the phytochrome superfamily (reviewed by [34]).

Some filamentous cyanobacteria such as *Fremyella diplosiphon* (also called *Calothrix sp. PCC 7601* or *Tolypothrix* sp. *PCC 7610*) are predicted to encode up to 27 unique phytochrome superfamily photoreceptors, mostly of unknown function [35]. *F. diplosiphon* has been, however, well studied regarding a process called complementary chromatic acclimation (CCA) [36,37]. When the organism is grown under green light, it changes the composition of its light harvesting antennas, the phycobilisomes, by synthesizing phycoerythrin. Under red light, the pigment phycocyanin is introduced instead. This allows the organism to fine-tune its photosynthetic machinery and to generate sufficient energy for growth under changing light conditions. The photoreceptor that is linked with the pigment change was identified as the sensor kinase RcaE [38,39]. *rcaE* deletion mutants *of F. diplosiphon* furthermore revealed the importance of RcaE for cell morphology and shape [40]. When grown under red light, cells show generally a more coccoid cell shape and trichomes are short, whereas under green light the filaments are long, and cells become rod-shaped [36]. This change is regulated by RcaE [40] through its regulatory effect on the expression of the transcriptional regulator BolA. Under red light, the auto-kinase activity of RcaE activity promotes the upregulation of *bolA* expression. BolA in turn binds to the promoter region of *mreB*, inhibiting transcription while under green light, lower levels of BolA result in an accumulation of MreB leading to rod-shaped cells [41]. *rcaE* deletion mutants were unable to change their morphology and remained coccoid under the different light conditions [40].

Phytochromes are also involved in phototaxis and motility, as it has been shown in *Synechocystis* sp. PCC 6803 (hereafter *Synechocystis*) (reviewed by [42]). In some filamentous cyanobacteria of the order *Nostocales* phototaxis is facilitated by the formation of motile hormogonia in response to the colour of light and its irradiance [43,44,45,46]. In *F. diploshiphon* for example, the formation of hormogonia is induced by red light while green light suppresses their induction [45,47]. However, hormogonia formation differs from other cyanobacterial differentiation processes such as heterocyst formation by the fact that it can be initiated by various other environmental cues, including nutrient concentrations and signals from symbiotic partners [48,49,50,51]. Once hormogonia formation has been initiated, a cascade of cellular development follows. This includes the synchronous division of cells, the fragmentation of trichomes at heterocyst-vegetative cell junctions and necridia (apoptotic cells), a reduction in cell volume and sometimes the formation of tapered filament termini [44]. We have shown in hormogonia from the branching cyanobacterium *Mastigocladus laminosus* SAG 4.84 that molecular exchange between neighboring cells is fast and that the reduction in cell volume could even further accelerate signal transduction to coordinate movement [52]. During gliding motility cell growth and division are arrested [53] but once the hormogonium reaches its destination, e.g., the host, cells elongate, divide and potentially differentiate into heterocysts. Various hosts have been identified for cyanobacteria, including bryophytes (hornworts, liverworts), the angiosperm *Gunnera*, the aquatic fern *Azolla*, fungi (forming lichens), the fungus *Geosiphon*, cycads and diatoms [54,55]. The cyanobiont (cyanobacterial symbiont) can either grow intracellularly or extracellularly in specialized compartments. For example in the case of the *Anabaena*-*Azolla* symbiosis the cyanobacterium resides in the leaf pockets of the fern [56]. Internal symbionts have additional challenges. Once inside the host, cell division needs to be regulated to avoid bursting of the cells. Indeed, it has been observed that the cyanobacterium *Calothrix rhizosoleniae* grows in shorter trichomes when inside the host cell than in its free-living state [57]. The signals and mechanisms behind the restricted growth remain basically unknown.

Although not further discussed in this review, cellular morphology can be changed by nutrient composition [58] as well as stresses such as temperature, salt, ultraviolet (UV) light and drought and are connected with the formation of ROS (reviewed by [59]). The aforementioned examples show the complexity and diversity of the environmental cues that influence morphology and cell shape in cyanobacteria. In the following paragraphs we will address the underlying genetic and structural components that are associated with the different morphologies.

### 2.2. Morphological Plasticity in Cyanobacteria

Morphological plasticity, or the ability of one cell to alternate between different shapes, is a common strategy of many bacteria in response to environmental changes or as part of their normal life cycle (reviewed by [60,61,62]). Bacteria may alter their shape by simpler transitions from rod to coccoid (and vice versa) as in *Escherichia coli* [63], by more complex transitions while establishing multicellularity (reviewed by [60]) or by the development of specialized cells, structures or appendages where the population presents a pleomorphic lifestyle [64]. The precise molecular circuits that govern those morphological changes are yet to be identified, however, a so-far constant factor is that the cell shape is determined by the rigid PG sacculus which consists of glycan strands crosslinked by peptides. To grow, cells must synthesize new PG while breaking down the existent polymer to insert the newly synthesized material. How cells grow and elongate has been extensively reviewed in model organisms of both, rod-shaped [65,66] and coccoid bacteria [67]. The molecular basis for morphological plasticity and pleomorphism in more complex bacteria, however, is slowly being elucidated as well (see a recent review by [62]). The protein complex responsible for cell wall elongation in rod-shaped bacteria is referred to as the elongasome and is composed of, among others, MreB, MreC, MreD, PBP2, PBP1A, RodA and RodZ [65,68,69]. MreB polymerizes into dynamic filaments that act as a scaffold ion which the PG synthesis machinery assembles [70,71]. MreB orchestrates elongasome assembly through interaction with transmembrane proteins, such as RodZ and MreC/D [65] and the direct involvement of MreB in cell wall morphogenesis was described upon the correlation between MreB polymers and PG deposition along the lateral cell wall [72,73,74].

The phylogenetic distribution of MreB seems to be ubiquitous in rod-shaped bacteria, which encode for at least one *mreB* homolog whereas, with few exceptions, coccoid bacteria lack *mreB*, supporting the theory that coccoid bacteria evolved from rods [75]. In fact, out of 253 sequenced bacterial genomes representing all possible shapes, 63% of the transitions from rod to coccoid were related to the loss of *mreB* [76]. In accordance with this hypothesis, an analysis of 141 fully sequenced cyanobacterial genomes found that only four species lack *mreB*. The four species included *Synechocystis*, *Crocosphaera watsonii* WH 8501, *Atelocyanobacterium thalassa* ALOHA and *Gloeocapsa* sp. PCC 73106, all of which are unicellular and coccoid. In all these cyanobacteria, the loss of *mreB* was accompanied by the loss of the complete *mreBCD* operon [77]. Among the myriad cyanobacterial shapes, all multicellular or baeocyte-forming cyanobacteria, independent of their cell shape, had a complete *mreBCD* operon, suggesting that the coccoid shape in some of those multicellular cyanobacteria is achieved by alternative mechanisms than simply a lack of MreB [77]. Few other cyanobacterial taxa have retained only a copy of *mreB* on their genomes and lack *mreC* and *mreD*. For example, the multicellular cyanobacterium *Trichodesmium erythraeum* contains only a partial *mreBCD* operon that lacks *mreD*, while the filamentous helical shaped *Arthrospira maxima* CS-328 only encodes for *mreB* [77]. Likewise, the helical shaped *Helicobacter pylori* contains both MreB and MreC but lacks MreD. However, *mreB* is not essential in this bacterium and is not involved in cell shape-determination [78]. Instead, a family of endopeptidases actively remodels and flexibilizes the PG crosslinks that enable the helical cell curvature needed for the successful colonization of the human stomach [79]. Likewise, the transition from helical to straight trichomes [80], could be governed by other proteins than MreB instead.

There are few studies that have tried to elucidate the role of MreB in cyanobacteria and even less that succeeded in obtaining *mreB* deletion mutants either partially or completely segregated. However, the common denominator of these studies is a function of MreB in cell shape maintenance, independent of the cyanobacterial morphology [81,82,83,84]. In the rod-shaped *Synechococcus elongatus* PCC 7942 (hereafter *Synechococcus*) and *Synechococcus* sp. PCC 7002, MreB appears to be essential [81,82] as only partially segregated mutants could be obtained. In both *Synechococcus* species partial loss of *mreB* resulted in cell shape defects where cells became more coccoid (Figure 1). As a result of the polyploid nature of cyanobacteria and their asynchronous DNA replication (reviewed by [85]), most cyanobacterial studies on MreB have focused on elucidating the role of this protein in chromosome partitioning. Indeed, *mreB* knockdown mutants in *Synechococcus* show disarranged chromosomal replication origin (ori)-foci, suggesting that MreB is involved in chromosomal positioning [83]. However, the role of MreB in chromosomal positioning seems to be species-specific as chromosome partitioning was not affected in a *mreB Synechococcus* sp. PCC 7002 depletion mutant [82]. Furthermore, MreB seems to be involved in cellular compartmentalization in *Synechococcus* as *mreB* mutants show altered carboxysome placements. However, this effect is likely indirect in which the function of MreB in cell shape determination provides the necessary structural framework to organize carboxysomes [81]. A similar pleiotropic and indirect effect might also explain the alterations in chromosome positioning in the *Synechococcus mreB* knockdown mutant [83]. A notable exception of the essential nature of *mreB* is *Anabaena* where *mreB* affects cell shape but was found to be dispensable for cell viability with combined nitrogen and did not affect chromosome segregation or placement [84]. *Anabaena mreB*, *mreC* and *mreD* deletion mutants were all characterized by an alteration of cell size, regardless of the growth conditions [58]. In wild type *Anabaena,* single cells are longer than they are wide (in respect to the trichome growth axis). In the *mreB*, *mreC* and *mreD* mutants, however, cells became more coccoid and seemingly inverted their orientation within the trichome, being wider than long (Figure 1). MreB, MreC and MreD additionally affected the *Anabaena* trichome length, possibly through a strengthening of the septal cell wall, which was found to be increased in diameter in the three mutants [58].

Cells within the trichomes of the multiseriate and branching cyanobacteria *Fischerella muscicola* PCC 7414 may display rod, coccoid or tapered shapes while also differing in cell size [19,93]. *F. muscicola* also shows alternative growth modes that include apical, septal, and lateral trichome growth, although it is still not known how MreB contributes to cell shape or PG synthesis in this cyanobacterium. Deletions of *mreB* could not be obtained in *F. muscicola*, but overexpression of GFP-MreB from the copper inducible *petE* promoter showed alternative MreB localization in the different cell morphotypes (hormogonia, young and mature trichomes) from *F. muscicola* [77]. Further assessment of *mreB* regulation and localization dynamics in the different morphotypes is necessary to elucidate the role of this protein in the morphogenesis of complex multicellular and branching cyanobacteria.

The above-described observations attribute a largely structural function to MreB in cyanobacteria, however, MreB has also been indicated to be involved in other cellular processes. *Spiroplasma eriocheiris*, a cell wall-less helical bacterium with swimming motility encodes for five MreB variants [94]. Together with the fibril protein, MreB was proposed to contribute to the propelling mechanism of *S. eriocheiris* by coordinating the length changes of their cytoskeletal ribbons [95]. Unlike any other cyanobacterium, some marine *Synechococcus* move by swimming using a still unidentified propulsion mechanism while surprisingly lacking apparent flagella systems [42]. Mechanisms such as the expulsion of a Newtonian fluid were excluded early on and instead a swimming mechanism was proposed to resemble the helical rotor mechanism propelling myxobacteria [96]. However, the involvement of MreB in cyanobacterial motility has so far not been demonstrated.

### 2.3. Different Modes of Cell Shape Regulation in Cyanobacteria

Despite their morphological complexity, cyanobacteria contain all conserved and so far known bacterial morphogens (Table 1). Understanding cyanobacterial morphogenesis is challenging, as there are numerous morphotypes among cyanobacterial taxa, which can also vary within a given strain during its life cycle [19]. Changes in cellular or even trichome morphologies are tasks that would require active cell wall remodeling and thus far no genes attributed to the different morphotypes have been identified in cyanobacteria [20]. Therefore, the most likely scenario is that genes or their products are differentially regulated during these cell morphology transitions [22], as it has been hypothesized for most bacteria [62]. In multicellular cyanobacteria, division of labor between cells within a trichome is achieved by different cell programing strategies. Thus, gene regulation occurs differentially in these specific cell types [30,97,98].

The multiplicity of mechanisms and life strategies displayed by cyanobacteria such as photosynthetic lifestyle, the presence of thylakoid membranes (with the exception of *Gloeobacter* [106]), carboxysome assembly, motility, nitrogen fixation and cell differentiation (i.e., hormogonia, akinetes, heterocysts and necridia) are associated with specific regulatory mechanisms that coordinate the different processes [30,97,98,107,108,109,110,111]. These regulatory mechanisms are complex and often intricate. Given this vast regulatory network, mutations that affect processes such as cell wall synthesis [26,101,105,112,113], intercellular transport [114,115] and cell division [88,90,116,117] may alter cell shape. Consequently, the function of the *mreB* gene or the entire *mreBCD* operon [58,84] was analyzed in gene overexpression or gene deletion mutants of these aforementioned processes (reviewed by [59]). In all studied cases, upregulation of *mreB* is associated with the transition from coccoid to rod shaped cells. MreB is also involved in the morphological transition during *N. punctiforme* hormogonia differentiation. Transcriptomic studies revealed that *mreB* and *rodA* were both upregulated in hormogonia from *N. punctiforme.* Similar *to M. laminosus* hormogonia [118], *N. punctiforme* hormogonia are characterized for having a smaller cell size and rod-shape in comparison with the larger and more coccoid cells of the mature trichome. Upregulation of *mreB* was also observed in a “branchless” morphotype of *F. muscicola* induced under sucrose supplementation [22], indicating that environmental growth conditions play a crucial role in cell shape regulation. Branch-less cultures were characterized by long trichomes that appear as nascent hormogonia previous to the detachment from the parent trichome [19,22]. Cells in the branchless cultures are longer, narrower and display a rod shape with tapered cells at the tip of the trichome compared to the more diverse cell morphologies (e.g., elliptical, rod-shaped, coccoid-shaped) in the parent trichome [22].

As photosynthetic microorganisms, iron has a pivotal role in cyanobacterial photosynthesis and defense against oxidative stress [119]. The transcriptional regulator FurA has been demonstrated as the master regulator of iron homeostasis in *Anabaena* [120] and was also shown to be involved in several other processes such as heterocyst differentiation and programmed cell death [119]. Overexpression of FurA in *Anabaena* lead to alterations in the cell shape, possibly through its positive regulatory function of the *mreBCD* operon [121]. Another prominent environmental factor affecting cyanobacterial cell shape is the availability of fixed nitrogen sources. Similar to the essential cell division gene *ftsZ* (discussed further below), *mreB* is differentially regulated during heterocyst formation [84]. Unlike *ftsZ*, *mreB* is upregulated during heterocyst formation in *Anabaena* pro-heterocysts [84,97] and an N-terminally GFP-tagged MreB localized to the cell poles in both vegetative cells and heterocysts [84]. The increase in MreB levels during heterocyst formation possibly provides the framework for the increase in cell size, which requires de-novo synthesis and integration of PG into the cell wall. In agreement with this, a recent study found that the incorporation of fluorescently labelled amino acids [122] into the *Anabaena* cell wall was elevated during heterocyst maturation [123]. PG biosynthesis enzymes, which are associated with the MreB-driven elongasome [68,124], were furthermore identified by several different groups to be essential for heterocyst formation [13,125,126], strengthening the importance of MreB function for heterocyst development. Additionally, *mreB* and *mreC* but not *mreD* are essential for diazotrophic growth of *Anabaena*, with a supposable function subsequent to heterocyst formation as *Anabaena mreB*, *mreC* and *mreD* mutants still differentiated heterocysts [58]. In the *Anabaena* wild type, cells are shorter during diazotrophic growth and longer in the presence of combined nitrogen [58]. This phenomenon can be explained by an increase in the levels of the global transcriptional regulator NtcA during diazotrophic growth. NtcA negatively regulates the *mreBCD* operon, leading to a reduced cell length. Consistently, an *ntcA* deletion mutant was characterized by an increased cell length [58].

Other factors that might regulate cell shape in cyanobacteria could be the interplay between the FtsZ and MreB cytoskeleton. In *E. coli* FtsZ and MreB can physically interact and this interaction is important for the progression from cell growth to cell division [127], whereas no direct effect of MreB on Z-ring placement and septum formation was observed in *Anabaena* [84]. This finding is in concert with the lack of interaction between MreB and FtsZ in *Anabaena* [77]. Notably, we recently showed that in the complex multicellular cyanobacteria *F. muscicola* and *Chlorogloeopsis fritschii* PCC 6912, MreB physically interacted with FtsZ [77], suggesting that their complex trichome and cell phenotypes could, in part, rely on the crosstalk between the elongasome and the FtsZ-driven divisome.

## 3. The Cyanobacterial Cell Division Complex—Function and Regulation

Numerous studies over the past years have conclusively shown that cyanobacteria not only possess a hybrid Gram phenotype in terms of their cell envelope but also possess proteinaceous structural determinants otherwise restricted to a single Gram type. The processes of PG and cell wall remodeling as well as cell septation rely on other divisome components that are recruited to the Z-ring. The Z-ring functions as a scaffolding structure for other divisome components but also potentially exerts constrictive force as indicated by FtsZ’s ability to bend liposomes [128,129]. In *E. coli*, more than 30 proteins have been identified as divisome or divisome-associated components, among those, 12 are essential and commonly associated with the divisome in the order: FtsZ → FtsA/ZipA → FtsE/FtsX → FtsK → FtsQ/FtsL/FtsB → FtsW/FtsI → FtsN (for reviews on bacterial cell division processes, please see [68,130,131]). The arrival of FtsN primes the divisome for septal PG synthesis and cell division. Homologs to some of those divisome proteins have been identified in cyanobacteria, including FtsE, FtsQ, FtsW and FtsI, while FtsA, ZipA, FtsL and FtsB are absent in cyanobacteria [28,86,88]. Other divisome-associated proteins from *E. coli* or *Bacillus subtilis* are likewise absent in cyanobacteria, including ZapA, ZapB, ZapC and EzrA. With one exception identified in *Synechocystis* [92], FtsN cannot be found in cyanobacteria. Similarly, a FtsX and FtsK homolog was so far only identified in *Anabaena* [86,132]. In agreement with their enormous morphological diversity, several morphological determinants specific to the Cyanobacteria phylum were also described, among those Ftn2 (ZipN) and Ftn6 (ZipS) [88,89,90,92,132,133,134]. Given the nonuniform nomenclature of cyanobacterial protein identifiers and to ease future research on cyanobacterial morphologies, we have collected a comprehensive list of important cyanobacterial structural determinants from the three widely used cyanobacteria *Anabaena*, *Synechocystis* and *Synechococcus* (Table 1). In the following sections, we will further elucidate the currently available information on some of those proteins, including their cellular context and functional properties.

### 3.1. Polymerization Properties of Cyanobacterial FtsZ

Cell division in bacteria is, with a few exceptions, strictly dependent on the function of the tubulin homolog FtsZ and its associated multiprotein complex, termed the divisome. FtsZ is an essential and highly conserved GTPase in almost all bacteria, Euryarchaeota, photosynthetic eukaryotes (i.e., in their chloroplasts) and even in some mitochondria [4,5,12,135,136,137]. Upon completion of chromosome segregation, FtsZ is the first protein to assemble at the future division site, forming a ring-like structure (the Z-ring) through GTP-dependent polymerization of FtsZ monomers into short protofilaments. Both, *Anabaena* and *Synechocystis* FtsZ contain the conserved glycine-rich GTP-binding domain, which is crucial for in vivo Z-ring formation and in vitro polymerization [88,138,139,140]. Unlike other bacterial FtsZ proteins, purified *Synechocystis* FtsZ assembles into a mixture of straight bundles, similar to chloroplast FtsZ, and toroidal filaments, indicating that the curved cyanobacterial FtsZ polymers could bend the cytoplasmic membrane [139,140]. Many cyanobacteria contain a highly variable N-terminal sequence extension of the FtsZ protein (between 20 and 80 amino acids long) that is absent in other bacteria but strikingly conserved among heterocystous cyanobacteria [88,140]. The N-terminal sequence is essential for *Anabaena* viability and, although FtsZ and an N-terminally truncated FtsZ (∆N-FtsZ) interact with each other, a ∆N-FtsZ-GFP fusion protein could not integrate into native Z-rings, possibly a result of its inability to interact with the FtsZ membrane anchor SepF [140,141]. While native *Anabaena* FtsZ forms toroids in vitro, ∆N-FtsZ only associates into filament bundles. As a consequence, the N-terminal peptide of *Anabaena* FtsZ, and possibly that of other cyanobacteria, likely promotes filament curling and decreases lateral filament bundling [140]. FtsZ filament curling or toroid-formation in *Anabaena* and *Synechocystis* FtsZ supports a constriction force of the cyanobacterial Z-ring itself, which is also described for *E. coli* FtsZ [139,142]. However, studies on *Prochlorococcus* Z-ring assembly suggest that it is likely not contractile in this species and possibly merely functions as a scaffold in oval-shaped cyanobacteria [143]. Straight filament bundles and toroids were previously reported for *E. coli* and *M. tuberculosis* FtsZ but only in the presence of crowding agents such as methylcellulose or polyvinyl alcohol. Given the larger cell diameter of *Synechocystis* and *Anabaena* (2–3 µm), filament bundling could be beneficial for their increased cell size compared to, for example, rod-shaped *E. coli* or *Synechococcus*, which are considerably smaller (1 µm cell diameter). Whether filament bundling exists in filamentous cyanobacteria with diameters less than 1 μm such as species of the genus *Halomicronema* [144] remains yet to be investigated. Many small rod-shaped bacteria also lack a signature motif in the H8 helix, which is likely responsible for filament bundling [139,140]. Consequently, cell shape possibly poses an evolutionary constraint on the functional diversification of proteins important for cell division. Considering the different cyanobacterial morphotypes, it will be interesting to test whether a similar observation also exists for processes that regulate cell growth in cyanobacteria, i.e., for the cell shape determining protein MreB.

### 3.2. FtsZ is Essential in Cyanobacteria

In cyanobacteria, *ftsZ* homologs were detected in every sequenced species and *ftsZ* was found to be essential in *Anabaena*, *Synechocystis* and *Synechococcus* [86,88,138,145]. Partial inactivation of *ftsZ* or addition of a tubulin assembly inhibitor (thiabendazole) causes cell filamentation (elongation) in the rod-shaped *E. coli* and *Synechococcus* and cell swelling in the coccoid *Synechocystis* [145]. Contrasting this, partial depletion of *ftsZ* or overexpression of the FtsZ assembly inhibitor *minC* results in a mixed filamentous/elongated and swollen cell shape in *Anabaena* [87,116] (Figure 1). Given that the ellipsoid cell shape of *Anabaena* can be considered a hybrid phenotype between the coccoid and rod-shape phenotype of *Synechocystis* and *Synechococcus*, formation of a hybrid cell shape defect upon impairment of cell division seems consistent. In the baeocytes-forming subsection II cyanobacterium *Chroococcidiopsis* sp. CCMEE 029, *ftsZ* is also essential and partial deletion disrupts the regularity in daughter cell arrangements, leading to cell aggregates. These aggregates, however, did not enlarge in cell volume compared to the wild type [146]. Therefore, the impact of impaired cell division appears to be highly dependent on the respective cyanobacterial morphotype and could result in different shapes in other so far understudied cyanobacterial subsections.

### 3.3. Cellular Localization of FtsZ in Cyanobacteria

As in other bacteria, FtsZ localizes to the middle of the cell in cyanobacteria, forming the typical Z-ring structure [86,88,116]. Consecutive Z-rings from neighbouring cells in the *Anabaena* trichome align parallel to each other. Z-rings from deeply-constricted *Synechocystis* daughter cells, however, form Z-rings that are perpendicular to each other [116], reminiscent of what we observed for true-branching subsection V cyanobacteria [77]. Using photobleaching of cyanobacterial autofluorescence coupled to super-resolution microscopy (STORM; stochastic optical reconstruction microscopy) of the unicellular, coccoid-shaped *Prochlorococcus* sp. MED4, a lateral resolution of 10 nm of Z-ring assembly was achieved [143]. Liu and colleagues found that FtsZ rings all contained small gaps, being non-continuous assemblies, and that FtsZ first polymerizes into incomplete and then complete rings, resembling the observations from *E. coli*. Consequently, their study thus supports the so-called patchy band model, where FtsZ assembles into discontinuous strings during cell division in contrast to the lateral association model that states that FtsZ polymers interact laterally to assemble into a complete Z-ring [143]. Studying other cyanobacterial morphotypes could consequently shed more light onto the debate about the FtsZ polymerization mechanisms in bacteria and might even indicate different assembly properties based on different cell shapes.

### 3.4. Transcriptional and Posttranslational Control of Cell Division in Cyanobacteria

*E. coli ftsZ* is transcribed in an operon together with *ftsA* (absent in cyanobacteria) and *ftsQ* (i.e., the *ftsQAZ* operon), whereas no *ftsQZ* operon was observed in *Anabaena*, *Synechocystis* nor *Synechococcus*, where *ftsZ* is independently transcribed from *ftsQ* instead [109,147], contrasting the identified *ftsQZ* operon structure in *M. aeruginosa* [148]. Given that cyanobacteria are photosynthetic organisms, it is not surprising that cell division and consequently *ftsZ* expression patterns are dependent on the circadian clock with an expression peak near dusk in *Synechococcus* and *Synechocystis* [147,149]. This circadian rhythmicity in *Synechococcus* is governed by the essential circadian clock protein kinase KaiC, through inhibition of Z-ring formation without impacting the cellular FtsZ protein levels [150]. In contrast, *ftsZ* transcription and cell division occur during the light cycle in the diazotrophic (nitrogen-fixing), unicellular *Cyanothece* sp. ATCC 51142 [151]. Diurnal control of *ftsZ* transcription also occurs in the marine, filamentous and nitrogen-fixing *T. erythraeum* IMS101. There, cell division and cell differentiation (into nitrogen-fixing diazocytes) occurs early during the dark period, which is preceded by an upregulation of *ftsZ* expression (and FtsZ protein level) [151]. Reminiscent of the dependency of cell division for heterocyst-development in *Anabaena* [152], diazocyte-development could be dependent on cell division in *T. erythraeum* [151]. Consequently, cell differentiation in cyanobacteria appears to be strongly connected to cell division and is halted in differentiated cells through a downregulation of *ftsZ* transcription and/or a decrease in FtsZ protein levels [116,153,154,155]. This notion is supported by the absence of Z-rings in terminally differentiated mature heterocysts in *Anabaena* [19,116,133]. Notably, loss of Z-rings precedes loss of detectable *ftsZ* transcripts, with the former taking place in immature heterocysts and the latter in mature heterocysts [133]. Thus, FtsZ is a cell division factor specific to vegetative cells in *Anabaena* [155]. The arrested cell division after heterocyst development is also apparent in true-branching filamentous cyanobacteria such as *M. laminosus* and other *Fischerella* species. During the life cycle the initially narrow trichomes with cylindrically shaped cells mature to wide trichomes with coccoid cells that give rise to true branches of cylindrical cells [52,156]. Once a heterocyst has formed within a certain type of trichome its morphology remains unchanged even when neighboring vegetative cells undergo the maturing process. An intriguing hypothesis is that in heterocysts, proteolytic FtsZ degradation is specifically increased, thus abolishing Z-ring formations (i.e., cell division). Although not shown to be specific to or increased in heterocysts, FtsZ-specific proteases were discovered in *Anabaena*, *F. muscicola* and *C. fritschii* cell extracts but not in the extracts of the non-cell differentiating *Synechocystis* or *E. coli* [77,157]. The precise nature of these proteases remains unknown but it was found to only cleave natively folded FtsZ in *Anabaena*, thus being structure and not sequence-specific [157]. Recently, PatA, a protein involved in the differentiation of intercalary heterocysts under nitrogen-deprived growth conditions and that localizes to the Z-ring and the cell poles [133], was found to function in destabilizing the Z-ring in *Anabaena* [133]. PatA interacts with ZipN and SepF, two crucial cyanobacterial cell division factors (discussed in more detail below) and it was hypothesized that this interaction ultimately promotes the loss of Z-ring structures during heterocysts development (Figure 1). Thus, PatA could be one component responsible for the loss of Z-ring structures in immature heterocysts, ultimately promoting cell differentiation progression [133]. In *T. erythraeum*, *ftsZ* transcription occurs only after DNA replication (extrapolated from *dnaA* gene expression) [151] as it was also shown in the unicellular, bloom-forming cyanobacterium *Microcystis aeruginosa* NIES298. There, *ftsZ* expression is repressed upon halt of DNA replication, suggesting that there are factors in the cell that sense the DNA content and regulate *ftsZ* transcription in response [148]. *Synechococcus* cells in stationary phase cultures rarely divide and elongate instead, largely due to the inhibition of DNA replication and consistent with a requirement of DNA biosynthesis for cell division [158]. Similar observations were also made for *Synechocystis* cultures in the stationary phase that revealed 4 to 10 times lower *ftsZ* transcript levels compared to log phase cultures; a mechanism that has been linked to cell density sensing [149]. In *Synechocystis*, two transcription factors (Sll0822 and Sll0359), which regulate *ftsZ* and *ftsQ* transcription, belong to the cyAbrB clade B of transcriptional regulators. Deletion of *sll0822* results in a cell division defect with swollen cells [159], similar to the *Synechocystis ftsZ* depletion strain [145]. In *Anabaena*, the cyAbrB transcriptional regulator CalA specifically regulates *ftsZ* expression in vegetative cells [160]. Whether the regulation of *ftsZ* expression by those transcription factors results in response to DNA content or other factors remains to be elucidated. Studies in *Anabaena* and *Synechococcus* have independently highlighted a positive correlation of DNA content with cell size and not with cell division, indicating that some cyanobacteria can sense their cell volume and adapt their chromosome content accordingly [116,161,162]. Consequently, it is conceivable that not FtsZ but MreB is indirectly involved in chromosome copy number determination, possibly through its regulatory function on cell shape and size. It is worth noting that the overall protein concentration within *Synechococcus* cells remained constant, regardless of the growth rate and is positively correlated with cell volume and DNA content [100].

### 3.5. FtsZ-Associated Regulators Control Cell Division in Cyanobacteria

In *E. coli* and *B. subtilis*, a number of factors, including the Zap proteins (ZapA/B/C/D) crosslink FtsZ polymers with each other or to the chromosome ends. The actin homolog FtsA, which, like FtsZ, is capable of filamentous assembly [163,164], and ZipA are major contributors of the first stage of Z-ring assembly. They anchor FtsZ to the cytoplasmic membrane through their interactions with FtsZ’s C-terminal peptide (CCTP) (reviewed by [165]). Both, ZipA and FtsA regulate divisome dynamics and recruit downstream divisome components to the Z-ring [166]. While FtsA is essential in *E. coli*, it can be deleted in *B. subtilis*, which results in filamentous cells that reveal a disturbed Z-ring formation [167]. Cyanobacteria lack ZipA, FtsA, ZapA and EzrA (a presumed Gram-positive-specific FtsZ membrane anchor [68]) homologs [86,148] and instead contain the cyanobacterial-specific protein ZipS (also known as Ftn6; [89]) and the cyanobacterial and plant-specific protein ZipN (also known as Ftn2) [92,132] (for a current model of the *Anabaena* divisome see Figure 2). Additionally, cyanobacteria possess SepF (also termed YlmF or Cdv2 [86]), a protein otherwise restricted to Gram-positive bacteria [168]. ZipN, ZipS and SepF all localize to the midcell in *Synechocystis* [88,90] suggesting that all three are important factors of the cell division machinery. They are, however, characterized by different levels of essentiality, depending on the respective cyanobacterial morphotype. *sepF* is essential in *Synechococcus* and *Synechocystis* [86,90], *zipN* is dispensable for *Synechococcus* but is essential in *Synechocystis* and *Anabaena* [88,89,132], whereas *zipS* can be deleted in *Synechococcus* and *Anabaena* but not in *Synechocystis* [89]. This inconsistency could reflect adaptations of the specific proteins to the respective morphology of its host.

SepF from *B. subtilis* is recruited early to the Z-ring and functions as a specific FtsZ membrane anchor, regulating the late septum formation [141,168]. In *E. coli*, FtsA or ZipA alone are sufficient to establish Z-ring anchorage to the cytoplasmic membrane, which is only lost upon simultaneous deletion of both [166]. The situation seems to be quite a bit more complex in cyanobacteria, possibly a result of the hybrid Gram phenotype and the morphological diversity. ZipN and its plant homolog ARC6 [169] contain a C-terminal transmembrane domain potentially suitable for membrane attachment [92,132], reminiscent of the amphipathic helix that mediates membrane localization of FtsA [165]. Furthermore, ZipN homologs contain a chaperone-like N-terminal DnaJ domain and a tetratricopeptide repeat (TPR) domain, suggesting that ZipN could function in mediating protein-protein interactions and/or affect protein folding [89]. Indeed, ZipN interacts with FtsZ in vitro and localizes to the Z-ring in *Synechocystis* and *Anabaena* (see Figure 2) [112,133], which is likely mediated by the DnaJ domain of ZipN, as removal of this domain results in diffuse cytoplasmic GFP-ZipN signals [112]. Reminiscent of FtsAs’ function in *E. coli*, ZipN is able to self-interact and functions as a *de novo* anchor of FtsZ to the cytoplasmic membrane in cyanobacteria [92,132]. Similar to *E. coli* FtsA, *B. subtilis* SepF assembles into round protein filaments and associates and bundles with FtsZ filaments in vitro [170]. *Synechocystis* SepF and ZipS directly interact with FtsZ filaments in vitro but only SepF is able to stimulate the assembly of FtsZ filaments [90]. Based on these observations, it was suggested that ZipS functions downstream of SepF, i.e., after the Z-ring is functionally assembled [90].

A direct function of SepF or ZipS in FtsZ membrane anchoring has not yet been described and depletion of *sepF* and *zipS* did not affect Z-ring formation but considerably altered the Z-ring structure and delayed cytokinesis, leading to swollen *Synechocystis* cells [90] (Figure 1). In the coccoid *Synechocystis*, *zipN* is required for normal cytokinesis and a *zipN*-depleted strain formed minicells or spiral-shaped cells [62] (see also Figure 1). Deletion of *zipN* and *zipS* but not depletion of *sepF* abrogates Z-ring formation in *Synechococcus*, leading to a patchy and diffuse pattern of FtsZ at the septum site [86], suggesting functional differences of *zipS* and *zipN* between the rod-shaped *Synechococcus* and the coccoid *Synechocystis*. Similar to the filamentous *B. subtilis sepF* mutant [168], rod-shaped *Synechococcus* cells deleted of *zipN* or *zipS* or depleted of *sepF* became filamentous or elongated [86,89]. Filamentous *Synechococcus zipN* or *zipS* mutants divided irregularly and can be up to 100 or 20 times longer than wild type *Synechococcus*, respectively [89]. Using light microscopy, both mutants appeared normal, however, ultrastructural analysis using scanning and transmission electron microscopy discovered that they are characterized by irregular cell bending and spiralization and have a decreased cell wall rigidity that is not a result of a PG layer defect [134]. None of this, however, affected the growth rate of the *zipN* and *zipS Synechococcus* mutants [89]. In the multicellular, ellipsoid-shaped *Anabaena*, *zipN* is essential while *zipS* can be deleted and both depletion of *zipN* or deletion of *zipS* lead to aberrant elongated and swollen *Anabaena* cells [89,132]. Thus, it seems that in those strains cell division but not cell growth is impaired. Confirming this, FtsZ was found to localize in a patchy and delocalized pattern around the cell division septa in the *zipN* and *zipS Synechococcus* mutants [86]. Likewise, depletion of *zipN* leads to a delocalization of FtsZ and a loss of Z-ring formation in *Anabaena*, implying a dysfunctional Z-ring assembly in strains lacking *zipN* [132].

The observation of swollen (*Synechocystis*, *Anabaena*) or filamentous (*Synechococcus*, *Anabaena* and *E. coli*) cells for the respective cell division mutants (for a depiction of several cyanobacterial cell division and cell shape mutant phenotypes see Figure 1) is considered to occur when cell septation is slowed down (or impaired) in relation to cell growth (i.e., lateral PG insertion) [4,17]. This idea is supported by a study that analyzed the proteome of filamentous *Synechococcus* cells deleted of *zipN* or *zipS*, which detected an upregulation of proteins involved in nucleotide biosynthesis like *dnaN* (DNA polymerase III beta subunit) and cell growth, including *mreB* [117]. As a result of the upregulation of cell-cycle-specific genes, the authors conclude that ZipN and ZipS likely act in a stage prior to cell division [117], which correlates with their occurrence early at the Z-ring during cell division in *Synechocystis* [88,90]. Notably, carboxysome-associated genes were also differentially expressed in the *Synechococcus zipN* and *zipS* mutants [117], being in concert with the observed decreased carboxysome count and the appearance of abnormal carboxysome-like structures not present in the wild type [134]. Consequently, both ZipN and ZipS have pleiotropic functions besides cell division and could be involved in carbon fixation. Given that carboxysome-segregation is dependent on McdA/B [171] and carboxysome subunit expression is affected by the deletion of *zipS* and *zipN* [117], a functional relationship between the cell division apparatus (i.e., divisome) and the carboxysome-segregation mechanism is possible and worth future investigation. More recently, ZipN was also attributed a function in cell differentiation in *Anabaena*. It was shown that ZipN protein levels, similar to FtsZ levels, are downregulated during heterocyst development, albeit at an earlier stage of heterocyst development (i.e., pre-heterocysts) [133]. Based on the interaction of PatA with ZipN (and SepF), the authors hypothesize that the initial binding of PatA to ZipN leads to a destabilization and loss of ZipN early during heterocyst-formation, followed by a destabilization and loss of FtsZ due to the lack of its membrane anchor (i.e., ZipN) in mature heterocysts. The subsequent downregulation of *ftsZ* transcription then seals the fate for the irreversible cell differentiation into heterocysts [133].

Highlighting its essential role for cyanobacterial viability and morphology, ZipN was found to interact with ZipS, SepF, FtsI and FtsQ in *Synechocystis* and *Anabaena* [92,112,132,172] with FtsK, FtsW, SepJ and SepI specifically in *Anabaena* [87,132,172] and with Cdv3 specifically in *Synechocystis* [92] (Figure 2). A more condensed summary of this and other known interaction networks of morphological determinants in *Synechocystis* and *Anabaena* is also given in Figure 3. Not much is currently known about the interaction profile of SepF and ZipS in cyanobacteria and unlike the FtsZ membrane-tethering function of ZipN, the precise function of SepF and ZipS remains to be elucidated. However, given that ZipS contains an N-terminal DnaD-like domain, which is involved in DNA binding, it could putatively act to bridge DNA replication with cell division in cyanobacteria [173]. It will be interesting to see whether this assumed function might provide the functional basis explaining the lack of a nucleoid occlusion system (explained in more detail below) in cyanobacteria.

### 3.6. The Divisome is Linked to the Sites of Cell-cell Connections in ANABAENA

Connection of multicellular cyanobacteria, including *Anabaena*, is, in part, mediated by an incomplete cell division and likely also through cell-cell-joining structures between neighboring cells (reviewed in [28,29,60]). The latter is indicated by a trichome fragmentation phenotype of mutants of the septal-localized proteins SepJ and SepI or septal junction proteins FraC and FraD [114,115,172]. Besides septal localization, SepJ, SepI and FraC additionally localize to the midcell in rings, reminiscent of the Z-ring [105,114,115,172,174,175], giving rise to the compelling connection between the cell division apparatus and the cell septa in *Anabaena*. This idea was followed up by numerous protein-protein interaction and fluorescent protein localization studies that revealed that, although SepJ does not interact with FtsZ, it does bind the cell division protein FtsQ [87], a bitopic membrane protein that putatively links the periplasmic to the cytoplasmic divisome proteins [176]. SepI also interacts with SepJ and with FtsI, SepF but not with FtsZ, ZipS, FraC, FraD, MreB, MinC and other Fts proteins (Figure 1) [172]. Unlike in unicellular bacteria, multicellularity in *Anabaena* is achieved through incomplete cleavage of the septal PG [177], hinting for a modified function and composition of the *Anabaena* divisome [87]. Consistent with a function of the divisome in septal junction integrity and thus *Anabaena* multicellularity, SepJ interacts with FtsQ and ZipN and its septal localization was largely lost in *ftsZ* and *zipN* depleted *Anabaena* strains, being mostly dispersed in patches [87,132]. Considering that FtsQ recruits numerous proteins to the *E. coli* divisome [178] and that ZipN interacts with an extensive set of divisome proteins [132], it is conceivable that both FtsQ and ZipN recruit SepJ to the *Anabaena* divisome, which then remains in the septa upon completed cell division [28,87]. Deletion of *sepI*, which, like *sepJ*, also functions in *Anabaena* multicellularity, nanopore formation and cell-cell communication did not alter Z-ring placement and only mildly affected cell shape and growth. These results suggest that SepI is a late divisome protein in *Anabaena* and its function is rather associated with septum integrity than divisome function [172]. Notably, and in contrast to the other characterized *Anabaena* divisome proteins, SepI was found to affect the colony morphology on agar plates. The implications of this for the growth of *Anabaena* in its natural habitat remain elusive.

### 3.7. Z-Ring—All in One Place

There are several main factors that function as negative regulators of and restrict Z-ring formation to the correct midcell placement [179]. They include the nucleoid occlusion (NO) system, consisting of SlmA in *E. coli* and Noc in *B. subtilis*, two chromosome-associated proteins that prevent Z-ring formation at sites occupied by DNA. Cyanobacteria lack SlmA and Noc homologs [17] and in *Synechococcus*, Z-rings were identified at sites occupied by chromosomal DNA, suggesting that cyanobacteria lack NO systems [86]. MipZ from *C. crescentus* belongs to the ParA/MinD family of ATPases and inhibits FtsZ assembly near the cell poles and the nucleoid through its association with the DNA-binding protein ParB [180]. Using bioinformatic searches, we could not identify MipZ homologs in *Anabaena*, *Synechococcus* and *Synechocystis*, consistent with the lack of a classical ParA/B/S-based DNA-segregation mechanism in cyanobacteria [83,171]. SulA, a FtsZ antagonist that sequesters FtsZ monomers and prevents Z-ring formation [181,182], is part of the SOS response as a reaction to DNA damage [179]. Homologs to SulA were identified in cyanobacteria ([86]. see also Table 1) and SulA was found to be essential in *Synechocystis.* Partial deletion of *sulA* resulted in cell division defects, prevented proper daughter cell segregation and led to cloverleaf-like cell aggregates [183]. *Anabaena* cells overexpressing *E.* coli but not *Anabaena* SulA became elongated, showed diminished Z-ring formations (like in *E. coli*), did not divide and, in accordance with an essential role of cell division for heterocyst-formation, did not differentiate heterocysts under nitrogen-deprived growth conditions [152]. The inability to differentiate heterocysts likely stems from the observation that *ftsZ* is downregulated 24 h after nitrogen-deprivation while *sulA* is upregulated. This argues for an additive effect to limit cell division through less FtsZ protein, which is additionally sequestered by SulA to prevent Z-ring formation [154]. An inhibition of Z-ring formation is likely also promoted through inhibition of FtsZ’s GTPase activity by SulA [152], which is essential for FtsZ filament-formation [184]. The Min system is the fourth known system that restricts the Z-ring to the midcell and works to prevent aberrant cell division planes in *E. coli* and *B. subtilis*. In *E. coli*, the Min system consists of three major proteins, MinC, MinD and MinE. MinC is the mechanistic antagonist to FtsZ polymerization through its interaction with the GTPase domain in FtsZ and concurrently with FtsZ’s CCTP domain, thus competing with FtsA to prevent membrane anchorage of FtsZ. MinC is recruited to the plasma membrane by its interaction with MinD, a Walker A-type ATPase and member of the ParA/MinD family. MinE, in the form of a MinE ring, associates to the membrane bound MinCD, causing its detachment from the membrane and giving rise to a spatiotemporal dynamic pole-to-pole oscillation of MinCDE, which is highest at the cell poles and lowest at the midcell. This gradient of the FtsZ inhibitor MinC ultimately restricts Z-ring formation at the correct midcell location (reviewed in [185]). *B. subtilis* does not contain a MinE homolog and instead possesses a coiled-coil protein called DivIVA, which localizes to areas of negative curvature—the cell poles or the division site of constricting cells—and recruits MinCD through a linker protein called MinJ. Consequently, no Min oscillation exists in the Gram-positive *B. subtilis* but Z-ring formation is statically inhibited by MinCD and DivIVA (reviewed in [68]). So far, the functional properties of the Min system were only elucidated in *Synechococcus* that, besides MinCDE, also contains a DivIVA-like protein called Cdv3. While one report suggested that Cdv3 is essential for *Synechococcus* [86], another report more recently reported the complete deletion of *cdv3* [91]. Although, the lack of functional domains essential for DivIVA function (e.g., membrane curvature sensing) in Cdv3 suggests that it is likely not a direct homolog of DivIVA [91]. Nonetheless, Cdv3 homologs, like DivIVA, are absent from other Gram-negative bacteria [86]. Given that *Synechococcus* contains MinCDE, essential for pole-to-pole oscillation, but also contains a sophisticated thylakoid membrane system that could potentially inhibit oscillation, it was unknown whether pole-to-pole oscillation can be recapitulated in cyanobacteria. MacCready and colleagues, however, elegantly modelled the existence of Min system oscillation in *Synechococcus* under the prerequisite that thylakoid membranes have a minimal permeability. They further showed robust Min system oscillation that spatiotemporally restricts Z-ring placement to the midcell in *Synechococcus*, demonstrating that the Min system can differentiate between the cytoplasmic membrane and the thylakoid systems [91]. Notably, they found two different modes that utilizes MinC’s ability to inhibit FtsZ polymerization: one dynamic, *E. coli*-like mode that employs dynamic MinC distribution through MinDE oscillation, although with a longer periodicity, and another, *B. subtilis*-like static mode in which Cdv3 and MinD recruit MinC rings adjacent to the Z-ring at the midcell position [91]. *Synechococcus* MinD is highly conserved and its C-terminal amphipathic helix but not the N-terminal ATPase domain is involved in membrane-targeting, while both are essential for MinD function [88]. MinE only shows low sequence similarity to *E. coli* MinE, and unlike in the *E. coli minB* operon, *minC* is not encoded together with *minDE* in *Synechococcus* [91]. Searching for MinD homologs, we identified one homolog in *Synechocystis* but two and three MinD homologs in *Synechococcus* and *Anabaena*, respectively (Table 1), raising the question of the function of the other MinD homologs. As in *E. coli*, *minE* is essential in *Synechococcus* [91], but non-essential in *Synechocystis* where deletion of *minE* has only a mild phenotype with rare minicell formations [88]. Although one study reported a fully segregated *Synechococcus minE* mutant with a 5′ inserted transposon, thus it is debatable whether *minE* is essential in *Synechococcus* [86]. Among the Min proteins, MinE likely functions as the essential regulator of Z-ring formation as *minE* depleted *Synechococcus* cells were filamentous and lacked clear Z-ring formation, which could still be observed in *minC* and *minD* knockout strains [91]. MinC overexpression induced cell enlargement (elongation in *Synechococcus* [91] and swelling in *Anabaena* [116]), and similar to SulA overexpression also halted cell division and cell differentiation in *Anabaena* [116], thus, attributing an important role of MinC in proper cell division and cell differentiation. Analogous to the situation in *E. coli*, *minC* and *minD* deletion causes defects in FtsZ placement resulting in a mixed population of minicells and elongated cells in *Synechococcus*. Analogous to this effect, a fully segregated *Synechococcus cdv3* mutant was also filamentous but did not form minicells [91]. In the coccoid *Synechocystis*, *minC*, *minD* and *minE* deletion strains did not enlarge in cell volume and instead became spiral-shaped (∆*minC* ∆*minD*) or formed minicells (∆*minC,* ∆*minD* and ∆*minE*) [88], whereas depletion of *cdv3* produced giant cells [92] (Figure 1). This suggests that in *Synechocystis*, Cdv3 could be of more importance for the control of cell division (i.e., inhibition of FtsZ polymerization, indicated by swollen cells) than the MinCDE system. Hence, while the deletion of the MinCDE pathway is possible, the Cdv3/MinD pathway is essential for *Synechocystis*; the exact opposite of the situation in *Synechococcus*. The most apparent difference between those two species is cell shape (coccoid vs. rod-shaped), consequently the different MinC-driven modes of FtsZ antagonism could be of different importance for different cyanobacterial cell morphotypes. It will be interesting to analyze the effect of the Min-system in *Anabaena*, which somewhat displays a hybrid morphotype between coccoid and rod-shaped.

Similar to MinC, Cdv3 in *Synechococcus* localizes to the midcell in rings [91], likely through an interaction with ZipN [92]. Overexpression of MinC and Cdv3 leads to the formation of remarkably long cell filaments, which can reach near millimeter-length for Cdv3-overexpressing strains [91,186]. As overexpression of MinC and Cdv3 does not inhibit cell growth, the increased sedimentation rates of those strains are now being exploited to optimize biomass harvesting procedures in cyanobacterial biotechnology [186]. Interestingly, low-light conditions or extended culture period (i.e., stationary phase cultures) are also associated with cell elongation in *Synechococcus*, leading to Min system-controlled asymmetric cell divisions [158]. The Min system enforces asymmetric division in elongated cells but ensures symmetric division in short daughter cells [187]. Notably, elongated cells produce more progeny cells than shorter ones and could act as storage units to overcome unfavorable conditions [158]. Considering all that information, it becomes apparent that cyanobacteria employ numerous mechanisms to regulate faithful cell division and utilize systems previously described to be restricted to either Gram-positive or Gram-negative bacteria. Finally, bearing in mind that many more cyanobacterial morphotypes have not yet been studied, it is conceivable that other, cyanobacterial-specific mechanisms to control cell division are yet to be discovered. Moreover, although a multitude of interactions have been identified among cell division/growth-related proteins, many other interactions are likely yet to be identified, making room to further explore the cyanobacterial cell division processes.

## 4. Coiled-Coil-Rich Proteins in Cyanobacteria

Despite relatively poor sequence conservation [188], eukaryotic intermediate filament (IF) proteins, the third major class of eukaryotic cytoskeletal proteins [2], reveal a robust tripartite building plan. IF proteins consist of highly variable N- and C-termini that flank a central α-helical rod-domain of conserved size (Figure 4). The rod domain consists of different coil segments that mediate the assembly into the characteristic coiled-coil (CC) structures with other IF proteins through lateral and longitudinal association, ultimately forming long IFs with a diameter of 11 nm (reviewed by [189]). About two decades ago, a functional involvement of an IF-like bacterial coiled-coil-rich protein (CCRP) in cell shape was described in the curved Gram-negative bacterium *C. crescentus* [9]. Although Crescentin is generally considered to be the first discovered bacterial IF-like cytoskeletal protein [190,191], the TlpA protein from *Salmonella enterica* was previously described as a bacterial CCRP with IF-like functions [192]. Nonetheless, Crescentin remains the best studied bacterial IF-like protein and has been shown to be essential for the typical crescent-like shape of *C. crescentus.* It aligns at the inner cell curvature [9], possibly mechanically controlling PG biosynthesis through a functional and potentially direct association with the MreB cytoskeleton [190]. Crescentin, reminiscent of eukaryotic IF proteins, forms filaments in vitro with a width of approximately 10 nm [9]. Although revealing compelling structural and domain similarities to eukaryotic IF proteins (Figure 4), given its restricted distribution to only one identified organism, Crescentin is considered to be likely no direct homologue of eukaryotic IF proteins but could rather be acquired by *C. crescentus* by horizontal gene transfer or as a result of convergent evolution [193,194,195]. The convergent evolutionary theory is supported by the ongoing discoveries of unrelated but structurally similar bacterial CCRPs that reveal IF-like characteristics. These proteins were shown to be involved in numerous different cellular functions, including cell shape (RsmP; [196]), cellular rigidity and polar PG biosynthesis (FilP, Scy and DivIVA; [197,198,199]), chemotaxis (Scc; [200]), gliding motility (AglZ; [201]), swimming motility and cell shape (*Helicobacter pylori* Ccrps; [78,202]), reminiscent of their eukaryotic counterparts (reviewed for example by [3,203]).

Given their seemingly ubiquitous involvement in cell shape, we recently searched for cyanobacterial CCRPs [204] that could be functionally involved in the manifestation of the enormous morphological diversity in the Cyanobacteria phylum [19]. In this study, we found that CCRPs are more prevalent in multicellular filamentous cyanobacteria compared to unicellular species. A specific reduction in CCRP proportion was identified in the genomes of the marine Picocyanobacteria, which could coincide with their reduced genome sizes [205]. The intriguing observation of higher CCRP counts in more complex cyanobacteria could indicate that CCRPs, at least in part, are important for the establishment of sophisticated morphological features in cyanobacteria. In fact, several septal junction-associated proteins, which are essential for the multicellular phenotype in *Anabaena*, contain CC domains [28,172]. Using a streamlined approach to readily test several candidate CCRPs with a newly developed in vitro polymerization assay allowed us to detect four novel filament-forming CCRPs in cyanobacteria. In *Synechocystis*, Slr1301 (termed HmpF_Syn_) is a homologous protein to HmpF from *Nostoc punctiforme* [206], which, similar to its homolog, was found to be involved in *Synechocystis* twitching motility (as also previously identified by [207]), possibly through its interaction with the pilus ATPase PilB [204]. Despite its high CC content, HmpF_Syn_ did not assemble into IF-like polymers in vitro and in vivo, highlighting that the pure presence of many CC domains is not sufficient to predict IF-like properties. Another *Synechocystis* CCRP, Slr7083 is encoded on a plasmid (the large toxin-antitoxin plasmid pSYSA) similar to TlpA from *Salmonella enterica* [192]. In contrast to HmpF_Syn_, Slr7083 assembles into a honeycomb-like web of protein filaments in vitro and localizes circumferentially to the cell envelope. Slr7083 also affected the cellular motility of *Synechocystis* (although to a lesser extent) and as it directly interacted with HmpF_Syn_, both CCRPs could be involved in cellular motility [204], reminiscent of the *H. pylori* CCRPs that regulate swimming motility [202] and AglZ, which is involved in gliding motility [201]. We also showed that a protein specific to multicellular cyanobacteria, Fm7001, polymerizes into extremely stable filamentous sheets at 4.5 M urea, a concentration where the eukaryotic IF protein vimentin only exist as tetramers [208]. This incredibly strong self-association capacity could function in the manifestation and stabilization of the *F. muscicola* trichome phenotype [204]. Interestingly, we also showed for the first time that a bacterial tetratricopeptide repeat (TPR) protein, All4981 from *Anabaena*, assembles into filamentous structures in vivo and in vitro, while interacting with a number of S-layer proteins. Notably, no S-layer has been detected in *Anabaena* [209], and a deletion of *all4981* could not be obtained, hampering a functional dissection of All4981. In our study [204], we further found two *Synechococcus* CCRPs of which Synpcc7942_1139 (HmpF_Syc_), a *Synechococcus* homolog to HmpF, is essential and has a severe impact on colony morphology, a novel trait of prokaryotic CCRPs. This essential property is in contrast to the non-essentiality of HmpF and HmpF_Syn_, suggesting specific functional adaptations to *Synechococcus*. Both mutants of *hmpF_Syc_* and *synpcc7942_2039* (hereafter *syc2039*) resulted in an elongated cell morphotype, reminiscent of other cell division genes previously identified in *Synechococcus*, including *ftsZ*, *ftn2* (*zipN*), *ftn6* (*zipS*), *cikA*, *cdv1*, *cdv2*, *cdv3*, *clpX,* and *minE* [86,91,210], indicating an impact of HmpF_Syc_ and Syc2039 on cell division. However, the cell elongation effect was not as severe as in the *zipN* and *zipS* mutants [86]. While the localization of HmpF_Syc_ was largely inconclusive, Syc2039-GFP formed spindle-like filamentous structures within several cyanobacterial strains and in *E. coli*, suggesting a strong self-sufficient assembly property. Nonetheless, in vitro Syc2039 filaments were not observed. Instead, Syc2039 seemed to rather be involved in DNA segregation as cells lacking syc2039 revealed an altered DNA distribution within the cell, reminiscent of *Synechococcus* cells treated with thiabendazole, a tubulin assembly inhibitor [145]. Although membrane association of bacterial CCRPs was described before (reviewed by [190]), no bacterial CCRP containing a transmembrane domain has been identified so far (reviewed by [12]). Consequently, the existing N-terminal TDM in Syc2039 further suggests that Syc2039 does not itself form filaments but rather associates with another filamentous system in bacteria [204]. Collectively, the myriad of different functional properties of cyanobacterial CCRPs, including cell and colony shape, cell division, motility, DNA segregation, and trichome integrity provide an initial foundation for future studies on the impact of CCRPs on the morphological and functional diversification in cyanobacteria. The employed in vitro polymerization assay using an unspecific NHS-fluorescein dye proved to be a valuable tool to conveniently detect polymerizing proteins in vitro (for a list of polymerizing cyanobacterial proteins see Figure 4) and could facilitate the identification of other filamentous proteins. We are currently working on the additional characterization of several *Anabaena* CCRPs and initial results indicate that some of those CCRPs, including ZicK and ZacK, could be involved in the stabilization of the linear trichome phenotype in *Anabaena*, extending the known impact of CCRPs from cell shape to trichome shape [211]. Noteworthy, ZicK and ZacK were also observed to be strictly interdependent to form heteropolymers in vitro and in vivo, describing a novel trait of bacterial CCRPs [211].

## 5. Undescribed Filamentous Systems in Cyanobacteria

Despite our analysis at the gene level, several conventional electron microscopy studies reported tubular, possibly cytoskeletal features in diverse cyanobacteria; however, they always lacked a clear identification of their protein composition. These observations can be divided into two subclasses: Microtubule-like structures (10–22 nm in diameter, length up to more than 1 µm) or thinner (3–8 nm in diameter) and less rigid microfilaments. Already in the late 1960s, a study observed microtubule-resembling, ~300 nm long structures with a diameter of 15 nm in an uncharacterized *Synechococcus* strain [213]. Thin sections of *Nostoc* strains revealed an even more intricate arrangement of tubules, consisting of an amorphous, ~1 µm long base plate parallel to the septum and numerous microtubule-like filaments perpendicular to it, protruding towards the cell center [214]. A similar complex could be visualized in *Anabaena* ([215] and reviewed in [216]); however, the described architecture shows strong similarities to phycobilisome arrays bound to thylakoid membranes, which were recently resolved in a native state in *Synechocystis* with cryo-electron tomography (cryoET) [217]. In *Anabaena*, it was further speculated that microtubular filaments could be important for the positioning of carboxysomes [215], whereas, the reported striated microtubules and sleeve bodies (Figure 5A) more resemble a membranous compartment or vesicles if re-analyzed with today’s knowledge and to a lesser extent cytoskeletal features [215]. In contrast, the finding of tubular structures bound to the cytoplasmic membrane in two *Nostoc* strains is even more remarkable nowadays [218], as they show striking similarities to the phage tail-like apparatus of the bacterial type VI secretion system, which was structurally discovered more than 30 years later and has not yet been identified in cyanobacteria [219]. The second prominent observations, which were termed microfilaments, could be visualized only in *Anabaena* [215] and in *Cyanothece* [220]. These finer filaments were observed in all areas of the cytoplasm and similar findings were recently made in our lab (unpublished), after artefact free-thinning of frozen-hydrated *Anabaena* cells with cryo-focused ion beam (FIB) milling [221] and subsequent cryoET. Multiple 5 nm wide and >500 nm long filaments bundled up in the cytoplasm and a repetitive subunit every 5.5 nm was discernible (Figure 5B). No discrete anchoring towards a membrane was detectable, although one end often co-localized with a thylakoid membrane stack. In cross-sections, the filaments revealed a tight packing with a center to center spacing of 11 nm (Figure 5C). We could observe these filaments in ~2% of our tomograms (*n* > 500 tomograms), which does not reasonably allow the suggestion of a function. Nevertheless, these data show that the cyanobacterial cytoskeleton is not yet fully understood and an integrative, multi-scale approach, from molecular biology to near-native imaging techniques like cryoET, is crucial to elucidate its diverse functions.

## 6. Conclusions and Future Perspectives

Recent scientific advances in the field of cyanobacterial research have started to unravel the mysteries and the evolution of cyanobacterial multicellularity, the cell–cell communication in multicellular cyanobacteria as well as cyanobacterial motility and provided growing insight into the molecular mechanisms that govern cell division and growth. This led to the insight that cyanobacteria are not just a mixture of Gram-positive and Gram-negative bacteria based on their cell wall characteristics but also because they harbor cell division genes specific to both Gram types and additionally possess cyanobacteria-specific cell division genes. Especially in multicellular cyanobacteria, a direct interplay between cell division processes and the establishment of cell–cell communication and ultimately multicellularity seems to exist. In the future, it will be of interest to analyze whether the cell shape-determining and MreB-driven elongasome is also linked to the incomplete cell separation process and intercellular communication in multicellular cyanobacteria. However, even apart from multicellular cyanobacteria and as a result of their mixed Gram phenotypes, it is intriguing to speculate that unicellular cyanobacteria have also evolved cell elongation processes and thus, elongasome functions in an alternative way to the well described model systems such as *E. coli* and *B. subtilis*. Nonetheless, given that several cell division processes appear to be cell shape-specific rather than phylum-specific, general statements about the functional properties of single proteins will likely remain restricted to the species or subsection level. The recent identification of cyanobacterial CCRPs with the property to form filament-like structures and their seemingly diverse cellular roles suggests that, at least in part, these proteins, like in other bacteria, could contribute to the special phenotype of cyanobacteria. The characterization of cell growth, cell division, and cytoskeletal processes in cyanobacteria has just begun and will likely provide us with unique insights into those fascinating bacteria. Technical advances like artefact-free sample thinning with cryo-FIB milling combined with correlative cryo-light microscopy/electron tomography will further allow in vivo visualization of these cytoskeletal features in a native state. Given their enormous ecological importance as primary inventors of oxygenic photosynthesis and their increasing importance due to the emerging climate change, cyanobacteria will likely receive more and more attention in the future that will also allow us to better understand their molecular circuits and consequently their unique adaptation strategies to the vast habitats that cyanobacteria populate.

## Figures and Tables

**Figure 1 life-10-00355-f001:**
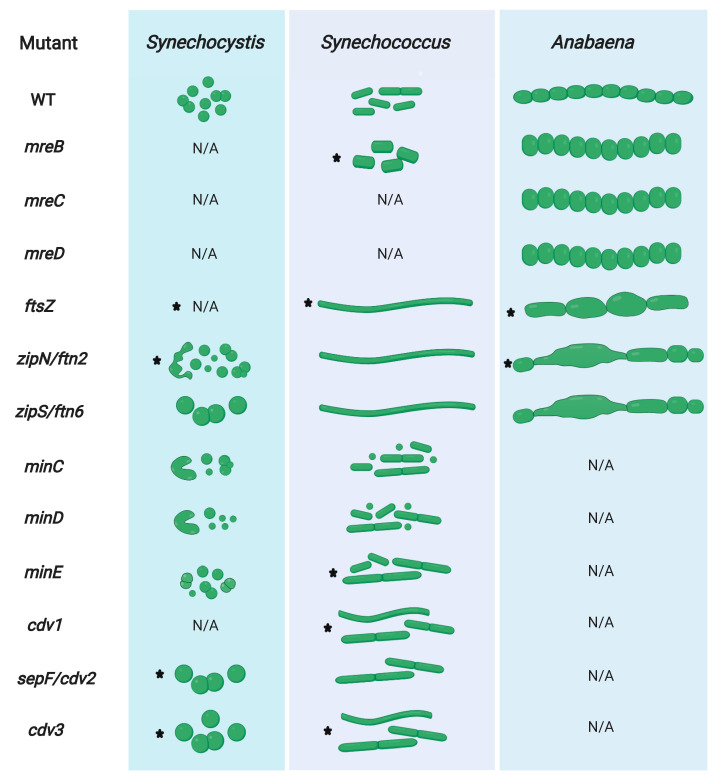
Cyanobacterial cell division and cell growth mutant phenotypes in *Synechocystis*, *Synechococcus*, and *Anabaena*. Stars indicate gene essentiality in the respective organism. Of note: while one gene can be essential in one cyanobacterial organism/morphotype, it does not necessarily mean it is essential in all other cyanobacteria. N/A indicates that no mutant phenotypes have been described. WT: wild type. Image created with BioRender.com. WT [19]; *mreB* [58,81,83,84]; *mreC* [58]; *mreD* [58]; *ftsZ* [86,87]; *zipN*/*ftn2* [86,88,89]; *zipS*/*ftn6* [86,89,90]; *minC* [88,91]; *minD* [88,91]; *minE* [88,91]; *cdv1* [86]; *sepF*/*cdv2* [86,90]; *cdv3* [86,92].

**Figure 2 life-10-00355-f002:**
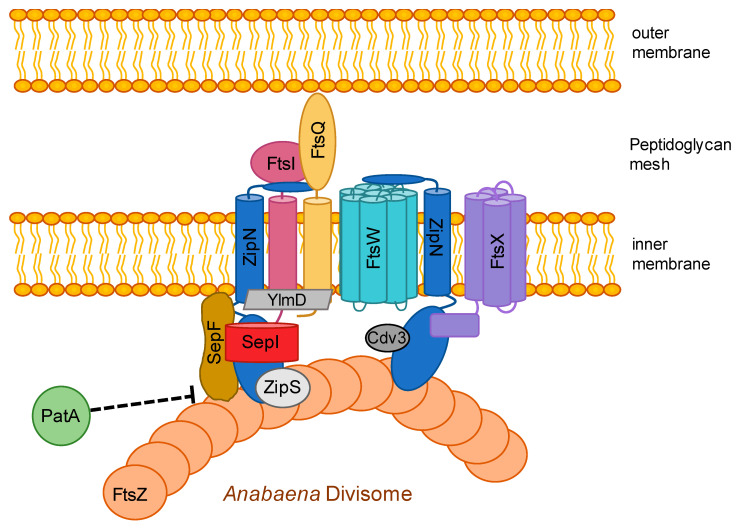
Proposed model of the *Anabaena* divisome. Proteins in grey shades are inferred from a previous model described for *Synechocystis* [92]. PatA is assumed to negatively interfere with the linkage of FtsZ to the cytoplasmic membrane through the loss of interaction with its presumed cytoplasmic membrane anchors SepF and ZipN.

**Figure 3 life-10-00355-f003:**
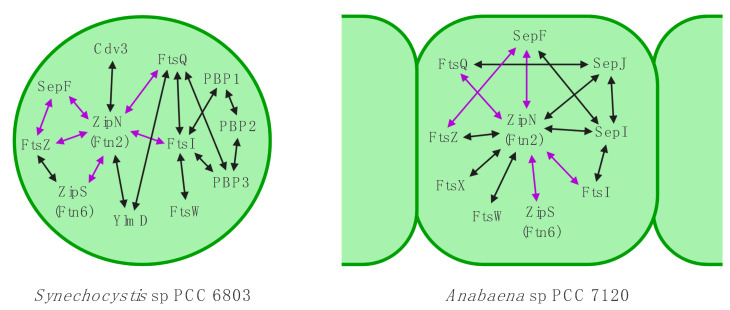
Interaction network of cell division/growth proteins in *Synechocystis* and *Anabaena.* Depiction of protein-protein interactions as identified by bacterial adenylate cyclase two hybrid assays and co-immunoprecipitation experiments. Interactions were identified in [87,88,90,92,132,172]. Black arrows indicate interactions solely found in one species so far, while purple arrows mark interactions found in both *Synechocystis* and *Anabaena*. Interactions only attributed to one species do not necessarily imply these interactions do not exist in the other but rather that these interactions were not yet tested for. Image created with BioRender.com.

**Figure 4 life-10-00355-f004:**
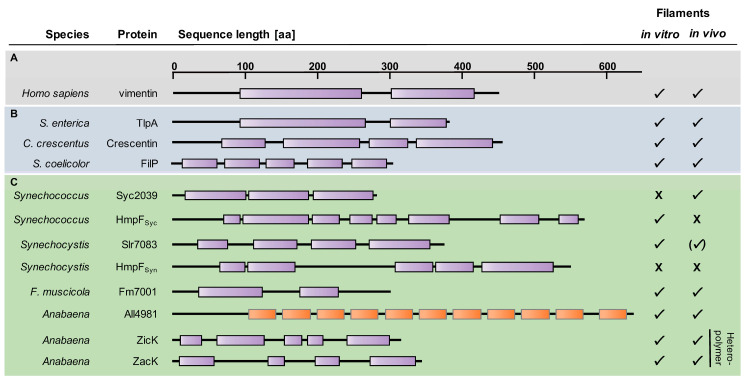
Bacterial coiled-coil-rich proteins. Depiction of coiled-coil-rich-regions (purple rectangles) in (**A**) the human vimentin, (**B**) previously described bacterial CCRPs, and (**C**) recently identified cyanobacterial CCRPs. Coiled-coil-rich regions were predicted using the COILS algorithm [212] or were obtained from [9,197,204,211]. Orange rectangles indicate TPR repeats that are also identified as coiled-coils by the COILS algorithm. Vimentin [188]; TlpA [192]; Crescentin [9]; FilP [197], Syc2039, HmpF_Syc_, Slr7083, HmpF_Syn_, Fm7001, All4981 [204]; ZicK, ZacK [211].

**Figure 5 life-10-00355-f005:**
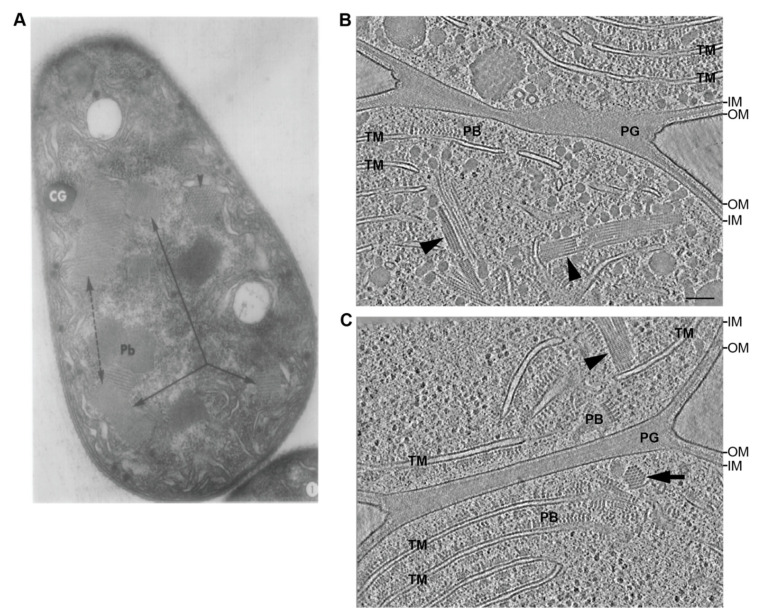
Electron micrographs of *Anabaena*. (**A**) Thin section of *Anabaena minutissima* showing striated microtubules (indicated by arrows). Pb, polyhedral bodies; CG, cyanophycin granule. (**B**,**C**) Cryo-electron tomograms of *Anabaena* showing uncharacterized intracellular filaments. (**B**) 5 nm wide filaments, with repetitive units every 5.5 nm, bundled together in the cytoplasm (arrowheads) and spanning parts of the cell. (**C**) Cross-section of these bundles revealed a tight packing with 11 nm spacing between filament centres (arrow). IM, inner membrane; OM, outer membrane; PB, phycobilisomes; PG, peptidoglycan; TM, thylakoid membrane. Bars, 100 nm. Shown are 13.5 nm thick slices. Figure 5A is reprinted from “The fine structure of striated microtubules and sleeve bodies in several species of *Anabaena*” [215], vol. 57, 1976, by Thomas E. Jensen and Robert P. Ayala with permission from Elsevier.

**Table 1 life-10-00355-t001:** Proteins involved in cyanobacterial cell division and cell morphology.

	Cyanobase Locus Tags and NCBI Accession Numbers			
Proteins	*Synechocystis*	*Synechococcus*	*Anabaena*	Function
FtsZ	Sll1633 (WP_010872126.1)	Synpcc7942_2378 (WP_011244037.1)	Alr3858 (WP_010997999.1)	Cell division
ZipN (Ftn2)	Sll0169 (WP_010873289.1)	Synpcc7942_1943 (WP_011244461.1)	All2707 (WP_010996860.1)	Cell division
ZipS (Ftn6)	Sll1939 (WP_010871735.1)	Synpcc7942_1707 (WP_011244694.1)	All1616 (BAB77982.1)	Cell division
Cdv1	Sll0227 (WP_010871341.1)	Synpcc7942_0653 (WP_011243187.1)	All4287 (BAB75986.1)	Cell division
SepF (Cdv2)	Slr2073 (WP_010872037.1)	Synpcc7942_2059 (WP_011378295.1)	Alr0487 (WP_010994663.1)	Cell division
Cdv3	Slr0848 (WP_010873766.1)	Synpcc7942_2006 (WP_011244399.1)	Alr4701 (WP_010998832.1)	Cell division
YlmD	Slr1573 (WP_010874196.1) ^a^	Synpcc7942_0346 (WP_011243479.1)	All5255 (WP_010999379.1)	Cell wall synthesis
YlmE	Slr0556 (WP_010874100.1)	Synpcc7942_2060 (WP_011244343.1)	Alr0486 (WP_010994662.1)	Unknown
YlmG	Ssr2142 (WP_010871471.1)	Synpcc7942_0477 (WP_011243354.1) ^b^	Asl2061 (WP_010996222.1)	Cell division
	Ssl0353 (WP_010873648.1)	Synpcc7942_2017 (WP_011244388.1)	Asl0940 (WP_010995114.1)	
YlmH	Sll1252 (WP_010872783.1) ^c^	Synpcc7942_1503 (WP_011378057.1) ^c^	Alr2890 (WP_010997041.1) ^c^	Unknown
MinC	Sll0288 (WP_010873891.1)	Synpcc7942_2001 (ABB58031.1)	Alr3455 (BAB75154.1)	Cell division
MinD	Sll0289 (WP_010873890.1)	Synpcc7942_0896 (WP_011242956.1) ^d^	Alr3456 (WP_010997606.1) ^e^	Cell division
	---	Synpcc7942_0220 (WP_011243604.1)	All2033 (WP_010996194.1)	
	---	---	All2797 (WP_010996948.1)	
MinE	Ssl0546 (WP_010873889.1)	Synpcc7942_0897 (WP_011242955.1)	Asr3457 (WP_010997607.1)	Cell division
SulA	Slr1223 (WP_014407090.1)	Synpcc7942_2477 (WP_011243937.1)	All2390 (WP_010996546.1)	Cell division
FtsE	Slr0544 (WP_010874063.1) ^f^	Synpcc7942_1414 (WP_011242455.1) ^f^	Alr1706 (BAB78072.1)	Cell division
FtsI	Sll1833 (WP_010871772.1)	Synpcc7942_0482 (WP_011243349.1)	Alr0718 (WP_010994893.1) ^g^	Cell division
FtsK/SpoIIIE	Sll0284 (WP_010873902.1) ^h^	Synpcc7942_0981 (WP_011242875.1) ^h^	Alr3799 (WP_010997940.1) ^h^	Cell division
	---	---	All7666 (WP_010993994.1) ^i^	
FtsN	Slr0702 (WP_010873961.1)	N/A	N/A	Cell division
FtsQ	Sll1632 (WP_010872127.1)	Synpcc7942_2377 (WP_011378434.1)	Alr3857 (WP_010997998.1)	Cell division
FtsW	Slr1267 (WP_010872891.1)	Synpcc7942_0324 (WP_011377535.1)	All0154 (WP_010994331.1)	Cell division
FtsX	N/A	N/A	All1757 (WP_010995925.1)	Cell division
CyDiv	N/A	N/A	All2320 (WP_010996476.1)	Cell division
SepI	N/A	N/A	Alr3364 (BAB75063.1) ^j^	Cell–cell contact
RodA	N/A	Synpcc7942_1104 (WP_011377865.1)	Alr0653 (WP_010994829.1)	Cell elongation
MreB	N/A	Synpcc7942_0300 (WP_011243524.1)	All0087 (BAB77611.1)	Cell elongation
MreC	N/A	Synpcc7942_0299 (WP_011243525.1)	All0086 (WP_010994263.1)	Cell elongation
MreD	N/A	Synpcc7942_0298 (ABB56330.1)	All0085 (BAB77609.1)	Cell elongation
BolA	Ssr3122 (WP_010871705.1)	Synpcc7942_1146 (ABB57176.1)	Asr0798 (WP_010994972.1)	Cell elongation
CikA	Slr1969 (WP_010872820.1)	Synpcc7942_0644 (WP_011243194.1)	All1688 (WP_010995857.1)	Circadian rhythm
PBP1	Sll0002 (WP_010873436.1)	Synpcc7942_2000 (WP_011378270.1)	Alr5101 (WP_010999227.1)	Cell wall synthesis
PBP2	Slr1710 (WP_010871874.1)	Synpcc7942_0785 (ABB56817.1)	Alr4579 (WP_010998711.1)	Cell wall synthesis
PBP3	Sll1434 (WP_010872930.1)	Synpcc7942_2571 (WP_011243849.1)	All2981 (WP_010997132.1)	Cell wall synthesis
PBP4	Sll1833 (WP_010871772.1)	Synpcc7942_0580 (WP_011377631.1)	Alr5326 (BAB77025.1)	Cell wall synthesis
PBP5	Slr0646 (WP_010873596.1)	Synpcc7942_1934 (ABB57964.1)	Alr5324 (WP_010999448.1)	Cell wall synthesis
PBP6	Sll1167 (WP_010872913.1)	Synpcc7942_0482 (WP_011243349.1)	All2981 (WP_010997132.1)	Cell wall synthesis
PBP7	Slr1924 (WP_010873199.1)	N/A	Alr5045 (WP_010999171.1)	Cell wall synthesis
PBP8	Slr0804 (WP_010872730.1)	N/A	Alr0718 (WP_010994893.1) ^g^	Cell wall synthesis
PBP9	N/A	N/A	Alr0153 (WP_010994330.1)	Cell wall synthesis
PBP10	N/A	N/A	Alr1666 (WP_010995835.1)	Cell wall synthesis
PBP11	N/A	N/A	Alr0054 (WP_010994231.1)	Cell wall synthesis
PBP12	N/A	N/A	All2656 (WP_010996812.1)	Cell wall synthesis

Absent in cyanobacteria according to [28,86,88]: FtsA, FtsB, ZapA, ZapB, ZapC, ZipA, EzrA, FtsB, FtsL, FtsN (although FtsN was reported in *Synechocystis* by [92]). a: Note that [92] identified Slr1593 as YlmD homolog, while we found this protein to be not the closest relative to YlmD from *Bacillus subtilis* or *Staphylococcus aureus* YlmD. b: YlmG as identified by [99]. c: predicted as photosystem II S4 domain protein. d: MinD identified by [100]; 27.5% sequence identity to Synpcc7942_0220. e: *Anabaena* MinD sequence identities: Alr3456+All2033: 23.1%; Alr3456+All2797: 25.9%; All2033+All2797: 59.4%. f: No FtsE was predicted in *Synechocystis* and *Synechococcus elongatus* according to [86]. g: Identified as FtsI by [101]. h: Predicted as YjgR family proteins of the HerA clade, relatives of FtsK [102]. i: Present on the *Anabaena* plasmid pCC7120beta. j: CyDiv is proposed to be part of an essential late divisome protein complex [103]. N/A: not available. **Note**: Differences in identified penicillin-binding proteins (PBPs) were found between [92,101,104,105]. Here, we present the data from [104] as it presents the most comprehensive analysis of cyanobacterial PBPs. “---” indicates absence of additional homologs.

## Data Availability

No new data were created or analyzed in this study. Data sharing is not applicable to this article.

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
