# Peer review of "Structural Determinants and Their Role in Cyanobacterial Morphogenesis"

_life, 2020, doi:10.3390/life10120355_

Round 1

Reviewer 1 Report

The manuscript of the review article “Structural determinants and their role in cyanobacterial morphogenesis” summarizes what is currently known about cell division, its regulation and determination of cell shape in cyanobacteria. This is an important topic, and this up-to-date manuscript will be very useful to everybody working with cyanobacteria or with cell division in any bacteria. I have few suggestions how to further improve the readability of the manuscript.

  1. Please add to Table 1 the biological function of each protein.
  2. A new figure showing cyanobacterial divisome structure would be very useful.
  3. Please also draw a cartoon showing steps in cell division.
  4. If possible, addition of figures showing phenotypes of cell division machinery mutants would be really nice.
  5. Text contains some repetition and could be also otherwise shortened.

Author Response

We would like to express our gratitude to the reviewers for their expedient comments during the performed peer-review. We are grateful for the possibility to strengthen our manuscript with the help of those comments as well for the opportunity to re-submit our research work. We have thoroughly revised our manuscript according to the reviewer’s comments and have added several new figures to ease the readability and comprehensibility of our manuscript.

In the following, please find our responses to the reviewer’s comments. Throughout the text, we have highlighted all deletions by crossed out red text and all text additions or new text components are indicated by green text.

Reviewer #1:

The manuscript of the review article “Structural determinants and their role in cyanobacterial morphogenesis” summarizes what is currently known about cell division, its regulation and determination of cell shape in cyanobacteria. This is an important topic, and this up-to-date manuscript will be very useful to everybody working with cyanobacteria or with cell division in any bacteria. I have few suggestions how to further improve the readability of the manuscript.

Comment 1: Please add to Table 1 the biological function of each protein.

Response 1: Table 1 now contains additional information about the proteins’ involvement in essential cellular processes, including cell division, cell elongation and cell wall synthesis/remodeling.

Comment 2: A new figure showing cyanobacterial divisome structure would be very useful.

Response 2: The requested figure is now provided as Fig. 2.

Comment 3: Please also draw a cartoon showing steps in cell division.

Response 3: The process of bacterial cell division has previously been elucidated and depicted by several excellent reviews. We have now added references to those reviews at the appropriate position in the text. However, too little is known about the respective single steps in cyanobacterial cell division and we therefore feel not comfortable presenting an illustration for which we have close to no experimental evidence for.

Comment 4: If possible, addition of figures showing phenotypes of cell division machinery mutants would be really nice.

Response 4: We fully agree with the reviewer and are happy to provide the requested figure as Fig. 1 in our revised manuscript.

Comment 5: Text contains some repetition and could be also otherwise shortened.

Response 5: We went through the text again and have made some considerable cuts specifically in the introduction.

Reviewer 2 Report

Comments to manuscript Life-1003741

Very interesting and important review. Meanwhile I suggest Authors to address the following issues for the final version of manuscript.

Section 1 (Introduction): contains a very high amount of introductory information, probably too much, sometimes it is difficult to follow the line of story.

Section 2/2.1., lines 145-147. “Cyanobacteria use a variety of chlorophylls…”- to my best knowledge, cyanobacteria contain only chlorophyll-a. I think this part should be thoroughly revised.

2.2. MreB the many functions of this protein are presented, but this part seems sometimes to be a collection of the information available to date, whit no coherent line of its nature and functions.

Presenting of MreB is a bit confusing throughout the manuscript. Please clarify this issue.

3.1. For a more comprehensive understanding of FtsZ functioning it would be great if we could see analogies with functioning of tubulin proteins.

3.4. control of FtsZ production, Line 506- “MreB is indirectly involved in chromosome copy number determination”- this issue is very interesting, more details should be given.

More Figures would be welcome to make the message more clear and help reader for a better guiding/ understanding of the paper. Because there are no sufficient Figures, reader is lost in a bulk of information. This is particularly true for the components and functions of the cyanobacterial cell division machinery:

  • a Figure (drawing) showing cyanobacterial cell division machinery including regulators of Z-ring assembly. The regulatory network presented on Fig. 1 is not sufficient for this.
  • it would be nice to present a Figure (scheme, drawing) on the division machinery and mechanisms of the lack of cell separation in filamentous multicellular cyanobacteria like Anabaena (see subsection 3.6.).
  • 2 is very important and spectacular. If possible, more TEM and/or CLSM images besides Fig.2 would also be beneficial to illustrate contents of text.

4. Coiled-coil-rich proteins in cyanobacteria as well as Section 5- these sections are written nicely and clearly, please follow this style for other sections too.

There are only a few typing/spelling errors, I indicated them in the annotated version of manuscript attached here.

Author Response

We would like to express our gratitude to the reviewers for their expedient comments during the performed peer-review. We are grateful for the possibility to strengthen our manuscript with the help of those comments as well for the opportunity to re-submit our research work. We have thoroughly revised our manuscript according to the reviewer’s comments and have added several new figures to ease the readability and comprehensibility of our manuscript.

In the following, please find our responses to the reviewer’s comments. Throughout the text, we have highlighted all deletions by crossed out red text and all text additions or new text components are indicated by green text.

Reviewer #2:

Very interesting and important review. Meanwhile I suggest Authors to address the following issues for the final version of manuscript.

Comment 1: Section 1 (Introduction): contains a very high amount of introductory information, probably too much, sometimes it is difficult to follow the line of story.

Response 1: We agree with the reviewer and have now reduced the introductory part by a considerable amount of text.

Comment 2: Section 2/2.1., lines 145-147. “Cyanobacteria use a variety of chlorophylls…”- to my best knowledge, cyanobacteria contain only chlorophyll-a. I think this part should be thoroughly revised.

Response 2:

The reviewer is correct that all cyanobacteria contain chlorophyll a. However, many cyanobacteria also contain additional chlorophylls such as chl b (e.g. in Prochlorothrix hollandica) and the recently discovered red-shifted chl d (i.e. mainly Acaryochloris marina) and chl f (many cyanobacteria; see e.g. Antonaru et al., 2020, ISME J). We have provided a review as reference.

Comment 3: 2.2. MreB the many functions of this protein are presented, but this part seems sometimes to be a collection of the information available to date, whit no coherent line of its nature and functions.

Response 3: We agree that the part about MreB does not provide a similarly comprehensive summary of its functional involvement in cyanobacterial morphogenesis as for example FtsZ can. This is, however, the result of the very few reports that have covered cyanobacterial MreB, precluding precise functional implications. We re-structured the MreB part to improve readability, though.

Comment 4: Presenting of MreB is a bit confusing throughout the manuscript. Please clarify this issue.

Response 4: Please see our response to comment 3.

Comment 5: 3.1. For a more comprehensive understanding of FtsZ functioning it would be great if we could see analogies with functioning of tubulin proteins.

Response 5: A comparison of FtsZ with the eukaryotic tubulin is not the aim of our review and we will therefore not go into more detail about their functional relationships. Several excellent papers have covered this relationship and we would like to refer the reviewer to those. For this, see for example: Wickstead and Gull, 2011, Journal of Cell Biology; Fink and Aylett, 2017 in Prokaryotic Cytoskeletons edited by Löwe and Amos; Erickson, 2007, Bioessays.

Comment 6: 3.4. control of FtsZ production, Line 506- “MreB is indirectly involved in chromosome copy number determination”- this issue is very interesting, more details should be given.

Response 6: We are afraid that not much more detail can be given at this point as this is still just a hypothesis based in the findings indicated in the preceding sentence. We have added a short addendum to the original sentence to explain the nature of this idea, though.

Comment 7: More Figures would be welcome to make the message more clear and help reader for a better guiding/ understanding of the paper. Because there are no sufficient Figures, reader is lost in a bulk of information. This is particularly true for the components and functions of the cyanobacterial cell division machinery:

Response 7: We agree with the reviewer and have now provided three additional figures and extended the former Fig. 2 (now Fig. 5).

Comment 8: a Figure (drawing) showing cyanobacterial cell division machinery including regulators of Z-ring assembly. The regulatory network presented on Fig. 1 is not sufficient for this.

Response 8: We have now included a figure depicting the presumed Anabaena divisome (Fig. 2).

Comment 9: it would be nice to present a Figure (scheme, drawing) on the division machinery and mechanisms of the lack of cell separation in filamentous multicellular cyanobacteria like Anabaena (see subsection 3.6.).

Response 9: The presumed and so far experimentally described division machinery of Anabaena is included in Fig. 2. Cell division and cell elongation mutant phenotypes of Synechocystis, Synechococcus and Anabaena are now depicted in Fig. 1. For a more detailed description of the underlying principles of cyanobacterial multicellularity, we would like to refer the reviewer to an excellent review by Flores and Herrero (“The multicellular nature of filamentous heterocyst-forming cyanobacteria”, 2010 in FEMS Microbiology Reviews) as this is not the aim of our review.

Comment 10: 2 is very important and spectacular. If possible, more TEM and/or CLSM images besides Fig.2 would also be beneficial to illustrate contents of text.

Response 10: We obtained permission to re-use one previously reported TEM picture for the referred figure and have now included it in our revised manuscript version. Other permissions were not granted free of charge and unfortunately, we do not have the financial resources to cover the costs for those permissions. Thus, we cannot further extend the former Fig. 2 (now Fig. 5).

Comment 11: 4. Coiled-coil-rich proteins in cyanobacteria as well as Section 5- these sections are written nicely and clearly, please follow this style for other sections too.

Response 11: We are happy about the appreciation of the last two sections of our review. However, as we have personally provided a bulk of the experimental information that fills those two chapters, we are naturally more aware of all the details of those findings, allowing us to describe them in more personal detail. As this is not the case for most of the other chapters, we can there mostly only report the findings of other groups, which of course cannot be done with the very same insight into the experimental matter.

Comment 12: There are only a few typing/spelling errors, I indicated them in the annotated version of manuscript attached here.

Response 12: We appreciate the reviewer’s attentiveness and have addressed the indicated mistakes.

Round 2

Reviewer 2 Report

Comments to revised manuscript Life-1003741

Very interesting and important review. Substantial and important revisions have been made. New Figs. 1 on division/growth mutants and Fig. 2 on the Anabaena divisome as well as Fig. 4 on bacterial coiled-coil rich proteins are great and they definitely make manuscript more readable and interesting.

Originally, I have stated: “Section 1 (Introduction): contains a very high amount of introductory information, probably too much, sometimes it is difficult to follow the line of story.” Regarding Response 1 of Authors: the clarification of Introduction section has been made.

There are two minor issues to be corrected, but these can be done even at the proof stage:

Line 25 of revised manuscript: please change “perquisites” to “prerequisites”.

Lines 90-91: Please delete this sentence, it is a repetition.

I find this review a very important contribution to the field worth to be published.